# Left Ventricular, Left Atrial and Right Ventricular Strain Modifications after Maximal Exercise in Elite Ski-Mountaineering Athletes: A Feasibility Speckle Tracking Study

**DOI:** 10.3390/ijerph192013153

**Published:** 2022-10-13

**Authors:** Paul Zimmermann, Max L. Eckstein, Othmar Moser, Isabelle Schöffl, Lukas Zimmermann, Volker Schöffl

**Affiliations:** 1Department of Cardiology, Klinikum Bamberg, 96049 Bamberg, Germany; 2Interdisciplinary Center of Sportsmedicine Bamberg, Klinikum Bamberg, 96049 Bamberg, Germany; 3Division of Exercise Physiology and Metabolism, BaySpo-Bayreuth Center of Sport Science, University of Bayreuth, 95440 Bayreuth, Germany; 4Department of Pediatric Cardiology, Friedrich-Alexander-University Erlangen-Nurnberg, 91054 Erlangen, Germany; 5School of Clinical and Applied Sciences, Leeds Beckett University, Leeds LS1 3HE, UK; 6Department of Orthopedic and Trauma Surgery, Friedrich-Alexander-University Erlangen-Nurnberg, 91054 Erlangen, Germany; 7Department of Orthopedic and Trauma Surgery, Klinikum Bamberg, 96049 Bamberg, Germany; 8Section of Wilderness Medicine, Department of Emergency Medicine at the University of Colorado School of Medicine, Denver, CO 80045, USA

**Keywords:** cardiopulmonary exercise testing, echocardiography, strain analysis, ski-mountaineering, training, athlete’s physique

## Abstract

Eleven world elite ski-mountaineering (Ski-Mo) athletes were evaluated for pronounced echocardiographic physiological remodeling as the primary aim of our feasibility speckle tracking study. In this context, sports-related cardiac remodeling was analyzed by performing two-dimensional echocardiography, including speckle tracking analysis of the left atrium (LA), right ventricle (RV) and left ventricular (LV) global longitudinal strain (LV-GLS) at rest and post-peak performance. The feasibility echocardiographic speckle tracking analysis was performed on eleven elite Ski-Mo athletes, which were obtained in 2022 during the annual medical examination. The obtained data of the professional Ski-Mo athletes (11 athletes, age: 18–26 years) were compared for different echocardiographic parameters at rest and post-exercise. Significant differences were found for LV-GLS mean (*p* = 0.0036) and phasic LA conduit strain pattern at rest and post-exercise (*p* = 0.0033). Furthermore, negative correlation between LV mass and LV-GLS (*p* = 0.0195, r = −0.69) and LV mass Index and LV-GLS (*p* = 0.0253, r = −0.66) at rest were elucidated. This descriptive reporting provided, for the first time, a sport-specific dynamic remodeling of an entire elite national team of the Ski-Mo athlete’s left heart and elucidated differences in the dynamic deformation pattern of the left heart.

## 1. Introduction

Ski mountaineering (Ski-Mo) has been accepted as a new Olympic sport for the 2026 Milan-Cortina Olympics [1]. Ski-Mo has been rated among the most strenuous endurance sports with the highest “hypoxic dose” (i.e., time spent in a hypoxic environment) [2,3,4,5], as it demands both maximal endurance performance during the individual races as well as high-intensity bouts during the sprint and vertical races in elite winter sports athletes [2,3,4,6,7,8,9].

In comparison to other winter endurance sports, e.g., Nordic Cross Country skiing or biathlon, Ski-Mo involves long uphill and downhill passages that have highly varying muscular demands. This leads to significantly different athletic features in comparison to other professional winter sports athletes [8]. Ski-Mo athletes tend to be smaller and younger than other elite endurance winter sports athletes [6,7,8]. In this context, significant physiological differences in morphological and functional cardiac remodeling could be identified for Ski-Mo athletes, especially due to left atrial (LA) remodeling, speckle tracking analysis of the left ventricle (LV), LV ejection fraction (LV-EF) assessment and LV Mass index [6,7,8].

The term athlete’s heart describes adaptions, namely physiological, functional and electro-physiological, caused by variable physiological sport-specific demands [8]. A conventional morphological and functional echocardiographic assessment might fail to distinguish between an athlete’s heart and controls, whereby functional strain rate evaluation provides further information for subclinical abnormalities and risk stratification [8,10,11,12,13,14,15,16,17,18,19]. These relatively new, non-invasive imaging techniques enhance the understanding of an athlete’s heart through a comprehensive characterization of anatomical and functional adaption, providing novel insights into the assessment of cardio-physiological adaption [20]. 

Especially the analyzed innovative echocardiographic data on pre- and post-exercise conditions with particular attention to the application of speckle tracking as dynamic functional assessment of cardiac remodeling might provide further insights into the characterization of biventricular and LA function in athletes and the impact of sport-specific cardiocirculatory functional adaption in these athletes [20]. 

The scientific evidence on the physiological aspects of competitive Ski-Mo athletes is still sparse. However, so far sport specific cardiopulmonary remodeling due to structural and functional adaption of the athlete’s heart has already been proven [6,7,8,21,22,23]. Under resting conditions sport specific structural remodeling, such as left atrial volume index (LAVI) and LV global longitudinal strain (LV-GLS), could be detected [7,8]. The objective of the present scientific analysis is to investigate the feasibility of biventricular and LA phasic deformation patterns after maximal exercise, as earlier studies solely focused on strain analysis at rest. In this context, the aim of this feasibility study is to investigate the correlations between strain pattern at rest and their dynamic changes post-exercise to enhance our understanding of an athlete’s heart. This comprehensive characterization of biventricular and LA functions might provide novel insights into the physiological adaption of an athlete’s heart, especially under environmental conditions, such as specific altitude training conditions [20,24]. Previous research revealed a complex sport-specific interaction between physiological response and altitude training, whereby hypoxia and training stress are combined [25]. Alterations of the autonomous regulation of the nervous system and subsequent modifications of the heart rate variability (HRV) have been studied before, differently depending on the training intensity at altitude [25,26,27,28]. These peculiarities might be promising to provide further interesting data to distinguish sport-specific cardiac remodeling in these elite world winter sports professionals, who often perform their sports in altitude, from balanced cardiomyopathies in the future [7,8,20]. 

## 2. Materials and Methods

The local ethics committee of the University of Bayreuth approved the study protocol (O1305/1-GB). The study was conducted in conformity with the declaration of Helsinki and Good Clinical Practice [29]. Before any study-related activities, our participating athletes were informed about the study protocol, and athletes were asked to give their written informed consent. 

### 2.1. Study Population 

Eleven young elite Ski-Mo professionals, all active members of the German National Team, participating in World championships and the World Cup season, were examined and evaluated in the preseason preparation time in summer during the season 2022. No participating athlete had to be excluded from the study due to post-COVID-19 infection syndromes. All participants were evaluated for anthropometric data, 12 lead electrocardiogram (ECG) and two-dimensional transthoracic echocardiography, including strain analysis at rest and post-exercise. The obtained data of the athletes (*n* = 11; male *n* = 8, female *n* = 3) were compared for different echocardiographic parameters at rest and post-exercise. All participating athletes did not have any medical history, and their sports-related history revealed no potential risk factors for sudden cardiac death in all athletes and within their families. 

All participating Ski-Mo athletes were winter sports professionals with a total amount of 20–30 training hours per week during high-volume training times and 5–10 training hours during low-volume training times. During low-volume training times, the athletes focus on continuous endurance training, such as running and cycling, as well as functional strength training and individual training to improve muscle disbalances [6]. An additional sample size estimation was not conducted since the included *n* = 11 represents the entire cohort of the German National Team competing at World Cup and World Championship levels. 

### 2.2. Echocardiographic Examination 

An echocardiographic functional and morphological assessment at rest and post-exercise using a commercially available echocardiographic system Phillips EPIQ 7 device with an X5-1 Matrix-array transducer (Phillips Healthcare, Eindhoven, The Netherlands), following a standard protocol, was performed [30]. In addition, heart rate and blood pressure were measured. Two-dimensional echocardiographic analyses were performed at rest following the general recommendations [8,30,31,32]. The systolic LV-EF was calculated using biplane Simpson rule, based on the apical two-chamber—as well as apical four-chamber view. Two-dimensional linear dimensions and LAVI were evaluated for both ventricles and both atria manually according to the recommendations [30,31,32,33]. An estimation of the RV systolic function at rest using the TAPSE (Tricuspide annular plane systolic excursion) was obtained in the apical four-chamber view. The LV Mass index and the relative wall thickness (RWT) of the LV were calculated using the formula recommended by the current guidelines [34]. We measured the pulse-wave Doppler in the apical four-chamber view referring to the peak early filling (E wave) and late diastolic filling (A wave) velocities to assess the LV diastolic function. Additionally, tissue Doppler imaging of the lateral mitral anulus in the apical four-chamber view was performed (peak early velocity E′) [30,31].

Furthermore, speckle tracking analysis of the athlete’s heart was recorded at rest as well as three minutes post maximal treadmill cardiopulmonary exercise testing (CPET) as post-exercise assessment, focusing on LV, RV and LA. In this context, we obtained the LV-GLS pattern, RV free wall longitudinal deformation (RV FW long.Def.), RV four chamber longitudinal deformation (RV 4C long.Def.) as well as phasic LA strain analysis by two-dimensional strain analysis in the apical views. The detailed phasic LA strain analysis was performed according to the recent European Association of Cardiovascular Imaging (EACVI) recommendations, including LA reservoir strain (LASr), LA conduit strain (LAScd) and LA contraction strain (LASct) assessment [35]. The LA strain assessment was not limited because no Ski-Mo athlete presented atrial fibrillation.

Each of the participating Ski-Mo athletes was evaluated for the prevalence of left and right heart valve regurgitation as part of the standard echocardiographic assessment [8,30,36].

### 2.3. Statistical Analyses 

Data were analyzed with Graph Pad Prism 8.2.1(279) (Graph Pad Software; San Diego, CA, USA). All data were tested for normal distribution via Shapiro–Wilk test. Data were tested for differences with paired *t*-tests, with statistical significance being accepted at *p* ≤ 0.05. All data are presented as mean ± SD. Pearson correlations were conducted for anatomical echocardiographic parameters as independent parameter with GLS, RV deformation pattern and LA reservoir, LA conduit and LA contractile at resting conditions as well as post-exercise in our participating elite Ski-Mo athletes. Due to the relatively small number of participating elite winter sports athletes, who were enrolled as the uniqueness of this reporting, we are not able to draw reliable conclusions by uni- and multivariate regression analyses, but we might point out interesting trends in the cohort of enrolled world elite winter sports professionals. 

## 3. Results

### 3.1. Baseline Athletes’ Characteristics and Echocardiographic Assessment at Rest

The baseline characteristics and anthropometric data of the participating male and female Ski-Mo athletes are presented in Table 1. The body surface areas were calculated with the Du Bois method.

In the two-dimensional echocardiographic assessment, all participating Ski-Mo athletes showed a low–normal to normal systolic LV-EF at rest estimated by the biplane Simpson method and did not show any relevant pathological regurgitation of the right and left heart valves. Only mild regurgitation at the tricuspid and mitral valves was revealed. There was no relevant systolic pulmonary artery pressure evaluated by tricuspid peak systolic velocity. The obtained LA and LV assessment did not differ in between the participating athletes. The baseline echocardiographic characteristics are presented in Table 2.

In comparison with data from sedentary control measurements presented by the German Society of Cardiology (DGK) [32] and with previous morphological and functional echocardiographic data from these athletes [7,8], the obtained sport specific morphological echocardiographic data [LVedd diameter, LV Mass index measurements, interventricular septal wall thickness at diastole (IVSd) and left ventricular posterior wall thickness at diastole (LVPWd), and systolic LV-EF] can be categorized to be in the normal range [7,8].

### 3.2. Speckle Tracking Analysis of the Right and Left Heart at Rest and Post-Exercise–Sport-Specific Functional Cardiac Remodeling

In the speckle tracking analysis, normal LV-GLS values at rest (−21.55 ± 3.44%) and slightly reduced values for LV-GLS post-exercise (−17.25 ± 3.39%) were elucidated. The difference between rest and post-exercise assessment was significant (*p* = 0.0036), as presented in Figure 1.

No significant differences were found for the RV free wall longitudinal deformation (RV FW long.Def. rest −28.17 ± 5.45% vs. RV FW long.Def. post-exercise −26.57 ± 6.24%, *p* = 0.48) nor for the RV apical four-chamber longitudinal deformation (RV 4C long.Def. rest −23.08 ± 2.09% vs. RV 4C long.Def. post-exercise −21.71 ± 4.89%, *p* = 0.40) at rest and post-exercise.

With respect to standard echocardiographic parameters, LA and LV geometric properties did not show significant interindividual differences. The evaluation of functional LA remodeling by the average phasic LA strain (LAS) during all three phases of the atrial cycle revealed significant differences in the comparison of resting and post-exercise conditions. In this context, across our participating athletes neither the LASr analysis at rest compared to post-exercise strain pattern (LASr rest 51.95 ± 11.55% vs. LASr post-exercise 43.92 ± 11.88%, *p* = 0.12) nor the LASct analysis at rest compared to post-exercise strain pattern (LASct rest −10.49 ± 7.45% vs. LASct post-exercise −15.46 ± 11.58%, *p* = 0.24) revealed significant differences. 

Significant differences could be elucidated for the LAScd analysis at rest versus post-exercise parameters (LAScd rest −41.45 ± 5.46% vs. LAScd post-exercise −28.45 ± 10.33%, *p* = 0.0033, as presented in Figure 1).

### 3.3. Sport-Specific Functional Cardiac Remodeling–Univariante Relationships between Morphological Echocardiographic Characteristics and Functional Remodeling as Speckle Tracking Analyses 

No significant correlations between LAVI and the athlete’s heart strain pattern could be revealed, whereby a negative correlation between LV mass and LV-GLS (*p* = 0.0195, r = −0.69) and LV-GLS and LV mass Index (*p* = 0.0253, r = −0.66) at rest could be proven, as presented in Figure 2.

## 4. Discussion

To the best of our knowledge, this is the first feasibility study to prove functional LA response due to exercise conditions in elite Ski-Mo athletes. These peculiarities provide novel insights into the assessment of Ski-mountaineers’ heart-to-exercise conditions at altitude and improve our understanding of physiological sport-specific demands in these winter sports professionals. Our previous research has shown that the heart of Ski-Mo athletes has adapted to physical exercise by individual sport-specific remodeling due to morphological and functional assessment [6,7,8]. Our additional novel presented results might provide further useful data to complete the understanding of the complex intraindividual Ski-Mo athlete’s cardio-physiological performance adaption and of their enhanced anaerobic capacity during race performance at altitude.

Cardiac and physiological adaptions due to exercise in elite athletes, mainly endurance athletes, represent a hot topic, and studies addressing this topic already exist [11,12,13,17,37]. The uniqueness of our study as a feasibility trial represents the performance evaluation of a small number of world elite winter sports athletes competing in one of the most strenuous endurance sports with the highest “hypoxic dose”, and their competitions are often performed in extreme altitudes, including vertical race components [3,4,6,7,8]. The detected strain rate measurements at rest for LV-GLS in our analyzed participating elite Ski-Mo athletes indicate normal values, despite mild differences compared to untrained controls and well-trained athletes of other sports [11,19,38,39]. However, up to now, LV strain assessment in sport professionals remain largely undefined and has to be interpreted in a multimodality approach assessing cardiac function following exercise [11,16,38,40,41]. Therefore, previous research revealed converse results with either progressive strain rate increase [15,42] or decreased LV-GLS and RV-GLS rates during exercise [42]. Our data elucidate normal LV-GLS strain rates at rest but a small decrease immediately post-exercise. Therefore, our study may serve as a pilot project to evaluate a Ski-Mo athlete’s heart due to altitude performance conditions. Previous research revealed the importance of energy storage during LV twisting to increase and to maintain stroke volume during exercise [24,43]. Prolonged strenuous variables, such as an athlete’s blood pressure response, variable preload conditions, heart rate response during exercise and exposure, especially to high-intensity interval training (HIIT) episodes at altitude, have to be considered in the interpretation and slight reduction of LV-GLS post-exercise in our Ski-Mo athletes’ data [44,45] and their potential negative impact on left and right ventricular function according to “exercise induced cardiac fatigue” [46,47,48].

RV longitudinal strain seems to provide an effective tool to assess sport-associated individual athletes’ adaption or alterations of the RV by exercise-related RV overload, even at an early subclinical stage [49,50,51,52,53,54,55]. Normal to slightly lower resting values of RV myocardial strain have to be shown to be a consequence of physiological cardiac remodeling rather than subclinical myocardial damage in elite endurance athletes [51,52,56,57,58]. No significant differences between pre- and post-exercise conditions in our small cohort could be proven. Although it might be difficult to derive reliable conclusions from our RV strain assessment, our descriptive reporting might contribute to a better understanding of sport-specific RV remodeling at altitude conditions. Several influencing environmental as well as physiological parameters, the impact of ethnicity, the beta-adrenergic desensitization, and the different levels of exposure to dynamic training on the RV, not all highlighted in our presented feasibility analyses, have to be taken into consideration for a focused and comprehensive assessment of right heart remodeling [48,59,60]. Next to the impact of the functional shift of RV as a novel marker of an athlete’s heart [48,49,61,62], we have to be aware that such RV remodeling might represent a proarrhythmogenic substrate in some highly trained athletes in the absence of known familiar predisposition-implicating the importance of long-term clinical assessment in the accompanying performance center [63,64,65].

Previous research demonstrated the importance of LA remodeling as a promising tool for the volumetric and functional characterization of the LA in athletes [66] as well as in patients with heart failure [67]. To the best of our knowledge, this is the first feasibility study to evaluate the acute LA response in Ski-Mo athletes. LA function, in general, has a great pathophysiological impact on the modulation of LV filling conditions as well as on LA response functions during exercise [67,68]. Our novel obtained data on exercise-induced LA adaption revealed significant differences for the phasic LAScd data at rest versus post-exercise values, whereas LASr and LASct strain patterns were not affected. The possibility to mobilize atrial functional “booster” capacity in response to exercise represents an important aspect of an athlete’s performance, whereby endurance training is known to be associated with an increase in venous return with a potential overstretching of the LA [67,69]. Although LA response according to a Frank–Starling mechanism up to a certain point has been described before [70], it has to be stated that LA response may vary to different exercise modalities, environmental conditions and loads [67]. Referring to this background, our observed LAScd reduction post-exercise might be interpreted due to high-intensity components mainly at altitude environmental conditions. Due to these extreme conditions causing hypoxia and modifications of the HRV [25,26,28], a certain impact on the specific Ski-Mo athlete’s LA response might be assumed. Functional LA properties have to be interpreted as a consequence of intense and chronic training in an athlete’s heart, as stated before [66]. Exercise-induced hypertension, variable preload conditions after endurance exercise and LVH are suggested to have the mean impact on alternated LAScd or the previously described LA intrinsic impairment [71]. Whether these LA remodeling persists during an athlete’s lifetime career or might be related to environmental conditions remains to be evaluated. Nevertheless, these alterations might be related to the higher risk of arrhythmias, especially atrial fibrillation (AF) known as paroxysmal AF, in young and middle-aged athletes (PAFIYAMA) [68,72]. Therefore, our feasibility study cannot appraise any chronological causalities on the LAS pattern and draw reliable conclusions, and further prospective research is necessary to strengthen the scientific evidence.

Sport-specific cardiac remodeling in extreme levels of endurance sports, such as Ski-Mo, is a previously known phenomenon displaying a continuous process with a broad spectrum of adaption during an athlete’s career [5,6,7,8,73,74,75,76]. Whether the LVH found in athletes is physiological adaption or a risk factor for the progression of an initial subclinical hypertrophy remains controversial in the literature [40,76]. Evaluating the impact of anatomical left heart remodeling on biventricular and LA deformation patterns at rest and post-exercise, we could reveal a negative correlation between higher LV Mass as well as LV Mass Index and lower LV-GLS strain values at rest in our elite Ski-Mo athletes. Due to the small sample size, data outliers might lead to a significant bias in the echocardiographic data assessment and consequently affect the correlation of LV mass index and LV-GLS. Although it is not possible to derive reliable conclusions from this small sample size of Ski-Mo professionals, our data assessment might be regarded as an additional descriptive characterization of an athlete’s altitude training-induced dynamic remodeling process, primarily appearing as a balanced process [76].

Our feasibility trial aims to elucidate sport-specific adaptions of elite Ski-Mo athletes, who compete as very endurant athletes with enhanced anaerobic capacity at altitude and under environmental influences. However, our study is not without limitations. Firstly, our measurements were acquired in the preseason preparation time in summer with respect to a certain deviation in individual training schedules. Secondly, the slightly anthropometric variability in the cohort of Ski-Mo athletes with a mixture of young and experienced athletes entails an intra-cohort variability and contributes to a certain standard deviation in our cardiac assessment, such as the LV Mass index. Furthermore, no comparable winter sports athletes with different ethnic backgrounds except Caucasian athletes were analyzed. Additionally, focusing on speckle tracking deformation pattern pre- and post-exercise, no complete echocardiographic examination and no calculation of average septal and lateral E’ data were obtained according to the current and latest recommendations [30,34]. During the treadmill CPET, no strain analyses were acquired because it was technically not feasible. Last, we were unable to draw reliable conclusions between echocardiographic assessment and parameters of CPET performance because the size of the entire elite German Ski-Mo team is small. This would not lead to statistically reliable conclusions. Future research might focus on a larger athlete’s sample size to verify and strengthen the scientific evidence base of the obtained findings of our feasibility reporting.

## 5. Conclusions

This report provides, for the first time, new evidence of sport-specific dynamic remodeling of the Ski-Mo athlete’s heart and elucidates differences at resting and post-exercise conditions in the strain pattern of the left heart against the sport-specific altitude training background of Ski-Mo athletes.

In conclusion, strain imaging is technically feasible in world elite Ski-Mo athletes and might contribute to a better understanding of sport-specific remodeling and probably enhanced physiological remodeling in these highly trained altitude athletes. Further data are warranted to characterize the sport-specific dynamic atrial and ventricular cardiac remodeling, and our obtained findings might pave the road to future studies with long-term follow-up and a greater number of athletes to verify and strengthen the scientific evidence base.

## Figures and Tables

**Figure 1 ijerph-19-13153-f001:**
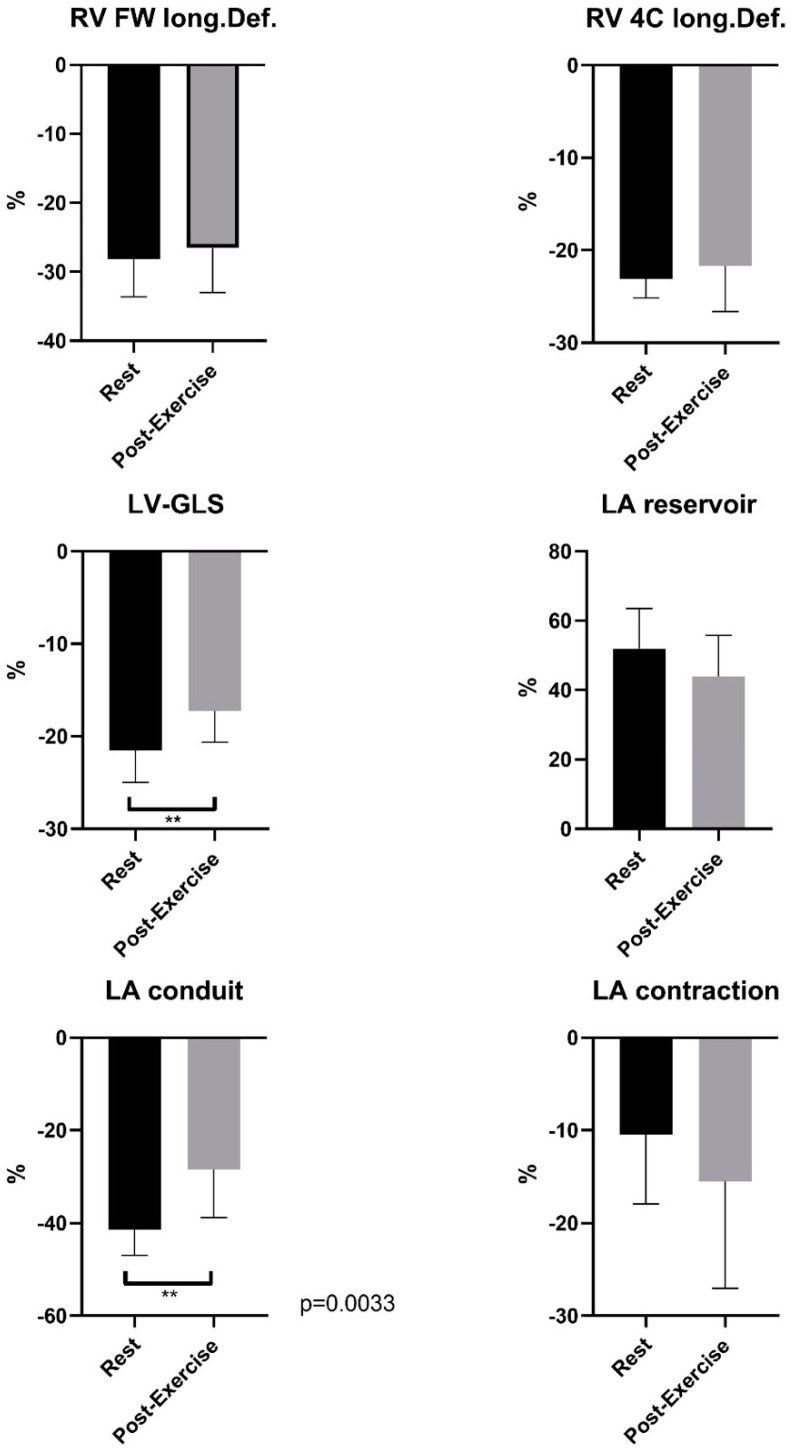
Speckle tracking analyses of world elite Ski-Mo professional–Comparison of mean LV-GLS, phasic LA strain and RV strain with significant differences between resting conditions and post-exercise. ** indicates *p* < 0.01.

**Figure 2 ijerph-19-13153-f002:**
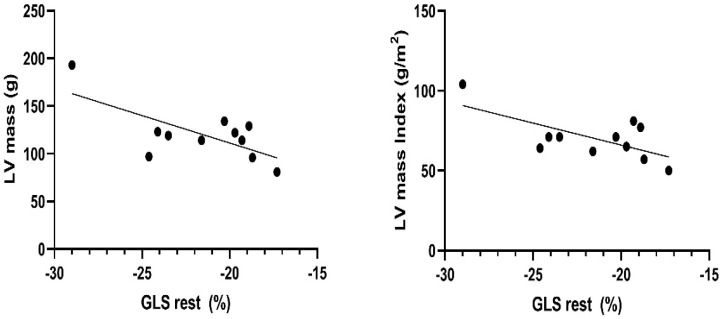
Negative correlation between LV mass and LV mass Index and LV-GLS at rest in world elite Ski-Mo professionals.

**Table 1 ijerph-19-13153-t001:** Baseline anthropometric Ski-Mo characteristics.

	Ski-Mo Male *n* = 8	Ski-Mo Female *n* = 3
**Age (years)**	20.5 ± 2.4	19 ± 0
**Height (cm)**	177.8 ± 4.6	164.0 ± 20.0
**Weight (kg)**	63.6 ± 6.1	52.1 ± 5.9
**BMI (kg/m^2^)**	20.1 ± 1.4	19.3 ± 0.4
**resting blood pressure systolic/diastolic (mmHg)**	119 ± 6.1 85 ± 3.2	110 ± 5.2 71 ± 2.3
**resting heart rate (bpm)**	41 ± 3.6	44 ± 2.5
**BSA (body surface area m^2^)**	1.79 ± 0.1	1.79 ± 0.1

Data are presented as a median with standard deviation. Abbreviations: cm, centimeter; kg, kilogram; m^2^, square meter; bpm; beats per minute. Bold: anthropometric athlete’s characteristics.

**Table 2 ijerph-19-13153-t002:** Baseline echocardiographic measurements (mean ± SD) of the Ski-mountaineering athletes and comparison to the stated control data of the position statement paper of the DGK (German Society of Cardiology, 2020).

	Ski-Mo Male *n* = 8	Ski-Mo Female *n* = 3	Reference Value Male	Reference Value Female
**LV edd (mm)**	47.13 ± 4.64	43.67 ± 2.31	42–58	38–52
**LV Mass Index (g/m^2^)**	71.38 ± 15.50	67.33 ± 12.35	49–115	43–95
**Relative wall Thickness RWT**	0.34 ± 0.05	0.37 ± 0.05		
**IVSd (mm)**	8.25 ± 1.28	7.33 ± 0.58	6–10	6–9
**LVPWs (mm)**	8.00 ± 0.93	8.00 ± 1.00	6–10	6–9
**E/A**	1.98 ± 0.32	1.90 ± 0.20		
**E/E′**	5.08 ± 2.18	5.53 ± 0.62		
**LAVI (mL/m^2^)**	28.13 ± 8.17	36.00 ± 3.00		
**RA (cm^2^)**	15.00 ± 2.39	14.00 ± 2.65		
**LV − EF_rest_ (%)**	61.38 ± 4.17	59.00 ± 1.00	52–72	54–72
**LV − EF_post-stress_ (%)**	70.00 ± 2.88	68.33 ± 0.58		

Data are presented as a median with standard deviation. Abbreviations: LV edd, left ventricle enddiastolic size; LV, left ventricular; IVSd, interventricular septal wall thickness at diastole; LVPWd, left ventricular posterior wall thickness at diastole; E/A and E/E′, parameters for diastolic function of the left ventricle; LAVI, left atrial volume index; RA, right atrium; LV − EF, left ventricular systolic ejection fraction. Bold: echocardiographic parameters.

## Data Availability

Individual anonymized data supporting the analyses of this study contained in this manuscript will be made available upon reasonable written request from researchers whose proposed use of data for a specific purpose has been approved.

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
