# Peer review of "Left Ventricular, Left Atrial and Right Ventricular Strain Modifications after Maximal Exercise in Elite Ski-Mountaineering Athletes: A Feasibility Speckle Tracking Study"

_ijerph, 2022, doi:10.3390/ijerph192013153_

Round 1
Reviewer 1 Report
This manuscript is well written and describes the novelty of a new echocardiographic approach on a sport specific population.
The strength of the paper is to set up a feasibility study on a new chapter of echocardiography based on endurance athletes, the authors well describe the aim and results. It is evident that they are experienced on this cardiological-sport specific field.
Nevertheless, I think there are some major issues to address before going forward:
- the utility of the research should be better addressed (especially the possibility to relate to sport and conditioning issue and to the real-world practitioner).
- an equally sex distribution is missing: can you add same number of male and female athletes? This represents an important theme in literature.
- the sample size is too small.
- In the limitation you can add the lack of other ethnicities studied.
- The relationship of these results and other endurance sports should be better addressed.
- Can you explain who performed the imaging? This is important to eventually stress the inter- and intra-operator variability of the technique.
Minors
Line 98 analysis
Line 158 please do not repeat date present in the table if you are not adding new concepts.
Author Response
Dear Editor,
Dear Reviewers,
Thank you very much for reviewing our manuscript entitled “Left ventricular, left atrial and right ventricular strain modifications after maximal exercise in elite Ski-Mountaineering athletes: a feasibility speckle tracking study”. Please find below an attached word file as point-to-point response to the specific comments.

Reviewer 2 Report
Zimmermann et al. have tried to investigate the feasibility of biventricular and LA phasic deformation pattern after maximal exercise. The aim of this study is to investigate the correlations between strain pattern at rest and their dynamic changes post-exercise to enhance our understanding of athlete´s heart.
Major concerns:
- small sample size
- as the authors have mentioned the study does not allow any further statistical analysis or conclusions due to the small sample size. Therefore, the study can only serve to describe certain circumstances or correlations.
- due to the small sample size data outliers might lead to a significant bias of the (echocardiographic) data, e.g. the moderate-to-good correlation between LV mass (index) and GLS due an athlete's LV mass of almost 200g, where as mean LV mass was approximately about 125g.
- the novelty of the study results is quite low. Many of the results can be seen as decriptive results of physiological adaptions after maximal exercise. Further, as mentioned in the discussion studies describing these changes do already exist, although they have not been performed in Ski-Mo athletes.
- The discussion is far too long. The authors should shorten the discussion significantly and focus more on the relevant results/conclusions/novel obsrvations. The introduction and methods can also be shortened.
Author Response

(The authors gave the same response as above.)

Reviewer 3 Report
Congratulations to the authors on this interesting manuscript. I have only minor comments:
- ine 49 cites systolic LV ejection fraction instead of LV ejection
- Regarding the echocardiographic examinations, calculate the RWT using the formula recommended by the current guidelines, i.e. (2x LVPW)/LV edd [PMID: 25559473]
- In the results section: include in the baseline athletes’ characteristics the body surface area (BSA) calculated with the Du Bois’ method; specify in the Figure 1 GLS as “LV-GLS” and LA contractile as “LA contraction”; Figure 2 is redundant with Figure 1, either exclude it or separate significant from non-significant differences between Figure 1 and 2
- Acknowledge in the limitation section that the echocardiographic examinations were not performed according to the current and latest Recommendations for Cardiac Chamber Quantification by ASE/EACVI [PMID: 25559473]; also calculations of an average of septal and lateral E' were not obtain even though it is recommended by the current guidelines and the ones cited in the study [PMID: 25559473, 18579482]
- Reading and citing this manuscript in the discussion can be helpful PMID: 35629111
Author Response

(The authors gave the same response as above.)

Reviewer 4 Report
The authors have done a good work on this feasibility speckle tracking study. I have some sort of questions . First of all , since the sample size is too small for data analysis , the authors have not discussed nor explained about the methodology of the study .
I also don't see the participants underlying/baseline criteria on medical history . I also don't see the strain speckle images of the participants . Then , we have a lot of method shows feasibility on speckle tracking study in both young and adult people . How this study has changed clinical practice or give any relevant clinical information for practice and diagnosis ?
The authors said this is the first report . But I have found the study on LV strain modification after maximal excercise in athelets with larger sample loads . (Santoro A, Alvino F, Antonelli G,at al . doi: 10.1111/echo.12791. Epub 2014 Nov 22. PMID: 25418356.
Author Response

(The authors gave the same response as above.)

Round 2
Reviewer 1 Report
Dear Authors,
I really appreciate your major revision, this work can improve the sports and mountain medicine and cardiology research, in particular related to endurance sports in extreme condition.
Kind regards
Reviewer 2 Report
Dear Reviewers,
I really appreciate the author's revision and their effort to improve the manuscript. Nevertheless, all major concerns were related to fundamental limitations and remarks (small sample size, lack of novelty and low quality of content).
Kind Regards.
Reviewer 4 Report
Thank you for the author's reply with the revised manuscript.
This newly revised manuscript looks much improved and has been discussed with more information. This short summary of the study findings with a comment on the findings is sounded.